



# Positive and negative influences of landfalling typhoons on tropospheric ozone over southern China

Zhixiong Chen[1], Jane Liu[1,2], Xugeng Cheng[1], Mengmiao Yang[1], Hong Wang[3]

[1] Key Laboratory for Humid Subtropical Eco-Geographical Processes of the Ministry of Education, School of Geographical Sciences, Fujian Normal University, Fuzhou, China
[2] Department of Geography and Planning, University of Toronto, Toronto, Ontario, Canada
[3] Fujian Meteorological Administration, Fuzhou, China

*Correspondence to*:  Jane Liu (janejj.liu@utoronto.ca)

**Abstract.** In this study, we use an ensemble of 17 landfalling typhoons over 2014-2018 to investigate the positive and negative influences of typhoons on tropospheric ozone over southern China. Referring to the proximity to typhoons and typhoon developmental stages, we found that surface ozone is enhanced when typhoons are 400-1500 km away during the initial stages of typhoons (e.g., from 1 day before and to 1 day after typhoon genesis). The positive ozone anomaly averagely reaches 10-20 ppbv at the daytime and 9 ppbv at nighttime compared with the background ozone level. Particularly, surface ozone at radial distances of 1100-1300 km is most significantly enhanced during these initial stages. As the typhoons approach and land in southern China, the influences of typhoons change from enhancing to reducing ozone. Surface ozone concentrations decrease with a negative ozone anomaly ranging between -12 % ~ -17 % relative to the background ozone level.  We explore the physical linkages between typhoons, meteorological conditions and ozone variations. Results show that during typhoon initial stages, the increasing temperature and weak winds in the atmospheric boundary layer (ABL) and dominating downward motions promote ozone production and accumulation over the outskirts of typhoons. While the deteriorating weather accompanied by dropping temperature, wind gales and convective activity reduces the production and accumulation of surface ozone when typhoons are making landfalling.

Variations of tropospheric ozone profiles during the differential developmental stages of landfalling typhoons are further examined to quantify the influences of typhoon-induced stratospheric intrusions on lower troposphere and surface ozone. Using temporally dense ozone vertical observations, we found two regions of high ozone concentrations separately located in the ABL and the middle-to-upper troposphere under the influences of typhoons. Averagely, the ozone enhancement maximizes around 10-12 ppbv at 1-1.5 km altitude at the typhoon initial stages. The ozone enhancement persists over a longer period in the middle-to-upper troposphere with a positive ozone anomaly of 10 ppbv at 7-8 km altitude shortly after typhoon genesis, and 30 ppbv near 12 km altitude when typhoons reach their maximum intensity. When typhoons are landing, a negative ozone anomaly appears and extends upward with a maximum ozone reduction of 14-18 ppbv at 5 km altitude and 20-25 ppbv at 11 km altitude. Though the overall tropospheric ozone is usually reduced during typhoon landfalling, we quantify that five of eight typhoon samples deduce ozone-rich air with the stratospheric origin (80 ppbv) above 4 km altitude, and in 3 typhoon





cases the ozone-rich air intrusions (60 ppbv) can sink to the ABL. This suggests that the typhoon-induced stratospheric ozone-rich air intrusions play an important role in surface ozone enhancement.

35



# 1 Introduction

It has been noticed that high ozone (O₃) episodes are frequently associated with tropical cyclones (TC) in the warm seasons over southern China (Huang et al., 2005; Lam et al., 2005; Jiang et al., 2008; Shu et al., 2016; Chow et al., 2018; Gao et al., 2020). Previous studies suggested that TCs often modulate meteorological conditions and hence alter photochemical production, accumulation, transport and dispersion of ozone. For example, when TCs approach, the fine and hot weather are associated with strong solar radiation and high temperature, and the overwhelming downward air motions are conducive to low wind speed and stable atmospheric boundary layer (ABL), all of which are responsible for high ozone episodes in the developed and populated Pearl River Delta (PRD) and Yangtze River Delta (YRD) regions (e.g., Shu et al., 2016; Zhan et al., 2020). *Still, it is not clear whether these findings of TCs' impacts based on individual cases, are applicable over large domains, e.g., both coastal regions and neighbouring inland provinces in southern China.* In recent years, rapid urbanization and economic development also take place in other regions of southern China in addition to PRD and YRD, and many cities suffer from continuous increases in ozone levels (Li et al., 2019). Also, southern China are frequently under the control of TCs. There are around 326 TCs formed over the western Pacific during 2000-2017 (Li et al. 2020), and averagely six typhoons make landfall annually in southern China (Zhang et al., 2013). Therefore, it is urgently needed to statistically investigate the influences of TC on tropospheric ozone over southern China, given the frequent TC activities from June to October and the close connections between TC and high ozone episodes.

Typhoon, also named hurricanes in the Atlantic and the eastern North Pacific, refers to the intensive kind of TC with maximum sustained wind speeds exceeding 37.2 m/s. Those typhoons that finally make their landfall in China raise more concerns due to their relatively larger sizes, higher severities, and more direct passages toward coastal regions and neighbouring inland provinces in southern China. *Though TCs have been regarded as one of the main synoptic patterns influencing surface ozone concentrations, a comprehensive understanding of ozone variations in space and time attributable to landfalling typhoons is lacking, as previous studies generally were limited to individual cases and regional domains and mostly focused on ozone enhancement only.* A typhoon circulation typifies a radius of $O(10^3 \text{ km})$ and can persist for several days with varied intensity and location that steers the ozone behaviours. Concerning the occurrences of ozone episodes and the spatiotemporal distribution and property of typhoons, Huang et al. (2006) found that when a typhoon is about 700-1000 km from the PRD, the region is already controlled by large-scale subsidence of typhoon and suffers high ozone. Roux et al. (2020) stated that typhoons at distances of 500-1000 km offshore provide a favourable environment for active photochemical reactions and hence high ozone episodes. It is also documented that surface ozone concentrations increase over southwestern Taiwan 2 to 4 days before the passage of typhoons (Hung and Lo, 2015). Recently, Zhan et al. (2020) found that in YRD ozone pollution episodes mainly occurred when a typhoon reaches the 24-h warning line (thick dashed line in Fig. 1) and the previous typhoon dies away in mainland China. *While it is of value to stress typhoon-induced ozone enhancement in the context of air pollution, the cleansing ozone associated with landfalling typhoons is also important for complete evaluations of typhoon influences on surface ozone concentrations and on long-term tropospheric ozone trends. Hence, a full insight into the*


*evolutionary influences of landfalling typhoons, i.e., both ozone enhancement and reduction effects, would further our understanding of the role and contribution of typhoons on surface ozone variations and tropospheric ozone.*

Typhoons consist of bands of convective clouds that can vertically penetrate into the tropopause region. Therefore, they can potentially perturbate the structure and chemical compositions in the tropopause region, and promote stratosphere-troposphere exchanges (STEs). For example, several studies show that typhoons can induce the downward intrusions of ozone-rich air (Jiang et al., 2015; Das et al., 2016; Li et al., 2018; Roux et al., 2020). Such intrusions can even reach the ABL and

deteriorate air quality there, as in the case of Typhoon Hagibis over southeastern coast of China reported by Jiang et al. (2015). However, some previous studies emphasize the role of typhoons in cleansing the air and reducing tropospheric ozone concentrations. They hold that the stratospheric intrusions of ozone-rich air are insignificant, and instead, the uplifting of marine ozone-poor airmass by typhoons decreases tropospheric ozone concentrations. In a recent study, Li et al. (2020) analysed 18-year ozonesonde measurements at a frequency of once per week over Hong Kong and Nara, and found that TC

including typhoons reduce ozone by ~20–60 ppbv from the mean near the tropopause. Noticing the positive and negative influences of typhoons on tropospheric ozone, researchers pointed out that such different influences are closely related to development stages and intensities of typhoons (e.g., Zou and Wu 2005; Midya et al., 2012). Therefore, given the evolving features of typhoons, sufficient ozone observations are necessary to adequately sample the fine-scale structure of ozone and hence quantitatively address the influence of typhoon-induced stratosphere intrusion to tropospheric ozone. Unfortunately,

few studies have been done because a large ensemble of typhoons and temporally dense ozone vertical observations are required to provide statistically reliable results.

In this study, we comprehensively investigate the successive response of ozone concentrations to landfalling typhoons over southern China. A large ensemble of landfalling typhoon cases over 2014-2018 is applied to examine the overall ozone behaviours and hence offer statistically reliable conclusions. The landfalling typhoons are divided into several developmental

stages to track their evolutionary features of location and intensity. Accordingly, the multiple impacts of typhoons on surface ozone variations, namely, the positive (enhancement) and negative (reduction) impacts during entire lifespan of typhoons are analysed and gauged. Given the importance of stratospheric intrusions to tropospheric ozone budget, the evolution of ozone profiles during landfall typhoons is analysed to quantify the contribution of external descending stratospheric ozone to lower tropospheric ozone. To realize this, temporally dense observations, including ground-based ozone and vertical ozone profiles

collected during typhoon seasons, are synchronized according to typhoon developmental features. Meteorological conditions are also analysed to reveal the physical linkages between typhoon evolutions and ozone variations in time and space. We intend to answer the following scientific questions:

*(1) How do surface ozone concentrations vary spatiotemporally under the influences of landfalling typhoons? What are meteorological factors responsible for such ozone variations?*

*(2) How the tropospheric ozone profiles respond to the differential developmental stages of landfalling typhoons? What are meteorological controls on the vertical ozone variations?*

*(3) Do the typhoon-induced stratospheric intrusions play a significant role in enhancing troposphere and surface ozone?*





The remaining paper is structured as follows. Section 2 describes the study domain and period, the ozone observations, meteorological data, and analysis methods. Section 3 presents the statistical distributions of surface ozone concentrations with
reference to typhoon developmental features. Section 4 shows the vertical variations of ozone concentrations during the evolutionary processes of landfalling typhoons based on temporally dense ozone profile observations. The impacts of typhoon-induced STE on the vertical ozone distributions is also presented. Section 5 offers the conclusions, discussions, and suggestions for future work.

## 2 Data and method

### 2.1 Typhoon data

The best track data of tropical cyclones for China is provided by the China Meteorological Center (CMA) (available at: http://tcdata.typhoon.org.cn/ zjljsjj_sm.html, last access: 30 May 2021) (Ying et al., 2014). The TCs can be classified into several categories according to their averaged wind speed, namely, tropical depression (TD, with wind speed of 10.8-17.1m/s), tropical storm (TS, with wind speed of 17.2-24.4 m/s), severe tropical storm (STS, with wind speed of 24.5-32.6 m/s), typhoon
(TY, with wind speed of 32.7-41.4 m/s m/s), severe typhoon (with wind speed of 41.5-50.9 m/s) and super typhoon (Super TY, with wind speed exceeding 51.0 m/s). We extracted the information about each typhoon over 2014-2018, including storm category, geolocation of cyclone centers (latitude and longitude), minimum sea level pressure, and maximum sustained wind speed. The information of best tracks is collected every 6 hours from 1949, and since 2017 it is updated to every 3 hours to better capture the typhoon evolution during its landfall.

The landfalling typhoons attract much attention due to their severity and direct passages toward densely populated lands. A total of 17 typhoons landed in China over 2014-2018 as shown in Fig. 1, which form a large ensemble for investigating impacts of typhoons on the overall ozone behaviours with high confidence. Given the evolving nature of typhoons with varied location and intensity, we divide typhoon development into several stages with reference to the timing of their genesis, first-time maximum intensity and landfalling. The genesis time (hereafter $T_g$) is obtained from the best track data when the typhoon
is first identified in TD category. Pre-typhoon periods are examined before 1 day ($T_g$-1d) and 2 days ($T_g$-2d) of $T_g$, and the conditions right after +1 day ($T_g$+1d) are also included in our study. The time of maximum intensity of typhoons ($T_{max}$) is determined when the maximum sustained wind speed peaks (also lowest minimum sea level pressure) for the first time. The time of landfalling ($T_{landing}$) is determined when typhoons first land in southern China and $T_{landing}$+1d represent the post-landfalling conditions +1 day after the typhoon landfall. Therefore, there are 7 developmental stages in total to represent
evolutionary characteristics of the entire typhoon lifespan. By sequence, they are $T_g$-2d, $T_g$-1d, $T_g$, $T_g$+1d, $T_{max}$, $T_{landing}$ and $T_{landing}$+1d. Ozone and meteorological conditions in the typhoon ensemble are synchronized according to the divided 7 stages and compared to illustrate how ozone varies with distance to typhoons at the different typhoon developmental stages in southern China.





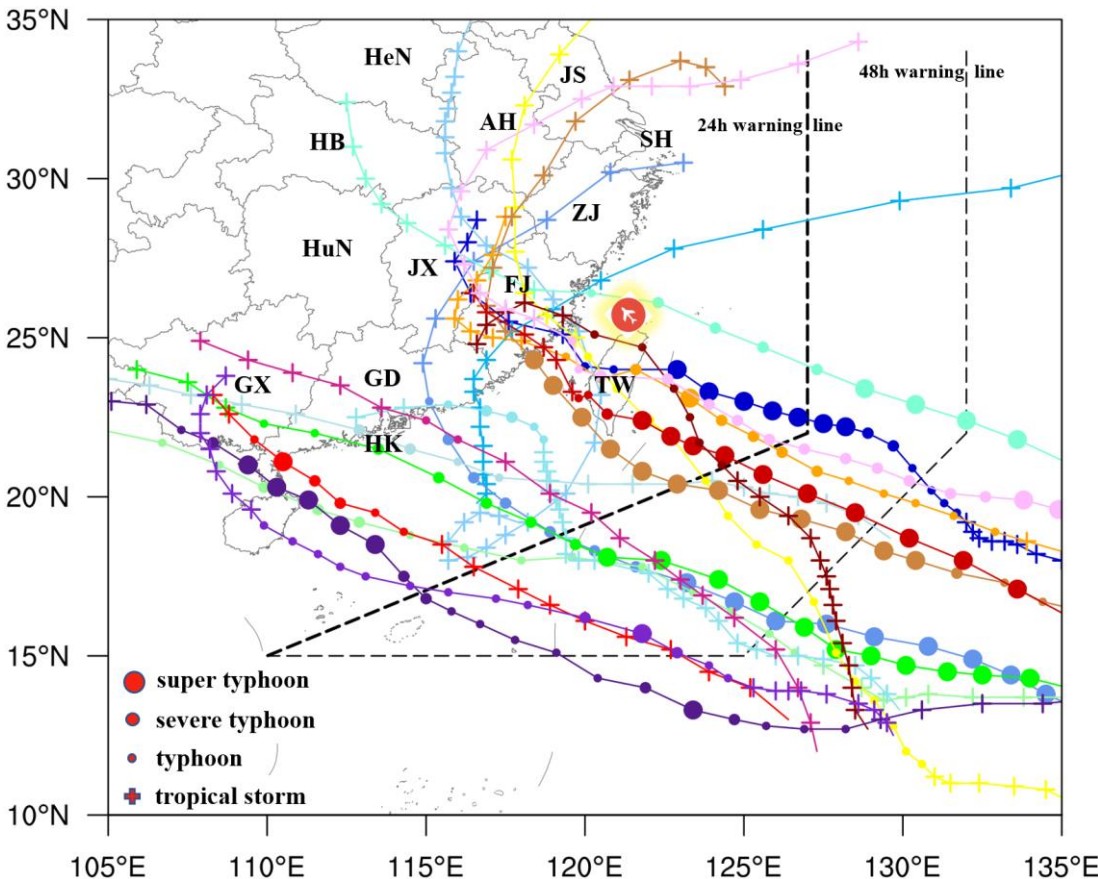

**Figure 1: The tracks (lines) and categories (symbols) of 17 landfalling typhoons over the western Pacific over 2014-2018. The thin and thick black dashed lines represent the 48-h and 24-h typhoon warning lines, respectively. The locations and boundaries for the provinces of interest are indicated by the corresponding abbreviations for Anhui (AH), Fujian (FJ), Guangdong (GD), Guangxi (GX), Henan (HeN), Hubei (HB), Hunan (HuN), Jiangsu (JS), Jiangxi (JX), and Zhejiang (ZJ). Shanghai (SH) district is also showed. The location, where vertical ozone profiles are available at Taoyuan International Airport (25.076 °N, 121.224 °E) of Taiwan (TW), is marked with a red plane symbol.**

**2.2 Ground-based ozone observations**

Routine measurements of surface air pollutants are provided by the China National Environmental Monitoring Centre. This nationwide network consists of more than 1500 stations distributed over 454 cities. Each station hourly measures six types of air pollutants including surface fine particles with an aerodynamic diameter of 2.5 μm (PM2.5) and of 10 μm (PM$_{10}$), ozone (O$_3$), carbon monoxide (CO), nitrogen dioxide (NO$_2$) and sulfur dioxide (SO$_2$) (Lu et al., 2018). The observational data are strictly quality controlled and can be accessed via a real-time air quality reporting website (http://106.37.208.233:20035/, last access: 30 May 2021).





This study focuses on the evolutionary impacts posed by typhoons on ozone concentrations over the coastal regions and neighbouring inland areas in southern China (Fig. 1). The hourly surface ozone concentrations in each city are calculated by averaging the observations from all the monitoring stations in that city. Two specific hours, the 1400 local standard time (LST, = 0600 UTC) and the 0200 LST (= 1800 UTC), are in coincidence with typhoon observation timing and used to represent typical ozone scenarios during the daytime and nighttime, respectively.

Provided with the geolocation of cities and typhoon centers, the radial distances between the ozone observation and the corresponding typhoon centers are calculated at each of the typhoon developmental stage for all the cities. Then these ozone concentrations are spatially averaged with a spacing of 200 km in radial direction. A background value of surface ozone concentrations ($T_{avg}$) is also calculated as the baseline by averaging the corresponding observations during the typhoon seasons (from June to October) over 2014-2018. Note in this calculation, the geolocation of typhoons at $T_g$ stage is used. For ozone features in pre-typhoon conditions at $T_g$-2d and $T_g$-1d, the geolocation of typhoon at stage $T_g$ is also applied as the typhoons are not generated yet. Ozone variations along the radial direction of each typhoon and at typhoon developmental stages are repeatedly calculated and averaged for taking mean ozone concentrations over the typhoon ensemble.

## 2.3 Airborne-based ozone observations

Airborne measurements of atmospheric chemical compounds are provided by the European Research Infrastructure program IAGOS (In-service Aircraft for a Global Observing System, https://www.iagos.org, last access: 30 May 2021) (Petzold et al., 2015). $O_3$, CO, nitrogen oxides ($NO_x$) as well as temperature, winds and relative humidity are measured by the in-situ sensors during flights around the world. For O3, it is measured by a dual-beam UV absorption monitor operated at 253.7 nm, and the concentrations are automatically corrected for pressure and temperature influences. The response time of $O_3$ measurement is 4 s, and the accuracy of ozone observations is estimated to be at ±2 ppbv (Thouret et al., 1998).

Sufficient IAGOS observations are collected during take-offs and landings at Taoyuan International Airport (25.076 °N, 121.224 °E, red plane symbol in Fig. 1) in Taiwan (TW). Considering data quality, the airborne observations associated with landfalling typhoons over 2014-2018 are screened and eight typhoons are selected. The ozone profiles are analysed with reference to the seven developmental stages of typhoons defined above. In the end, we obtain 234 profiles in total with 21 profiles for the genesis stage of typhoons and 34 for maximum stage and 29 for landfalling stage. These temporally dense ozone profiles guarantee to adequately resolve the successive response of ozone distribution to different typhoon stages, and hence provide a comprehensive insight into contributions of typhoons to tropospheric ozone budget. The ozone profile data are processed into a uniform 100-m vertical resolution by averaging observations over 100 m thick layers from 0 to 12 km above sea level. Same as the processing of surface ozone observations, the ozone profiles associated with each developmental stages of each typhoon are averaged over the typhoon ensemble. The vertical profiles of temperature and wind observations are processed in the same way.



### 2.4 Reanalysis meteorological data

The typhoon-induced meteorological influences on ozone are analysed using the MERRA-2 (The Modern-Era Retrospective Analysis for Research and Applications, Version 2) reanalysis data, which are produced by NASA's Global Modeling and Assimilation Office (GMAO, https://gmao.gsfc.nasa.gov/ reanalysis/MERRA-2, last access: 30 May 2021). The MERRA-2 data have a spatial resolution of 0.5 °× 0.625 ° and 72 vertical levels. The reanalysis data have been evaluated and found to match well with the observations from Chinese weather stations (Li et al., 2019). The gridded meteorological variables,

including temperature, wind, vertical velocity and potential vorticity (PV), are extracted from MERRA-2 during each of the typhoon developmental stages to investigate meteorological linkages to the production, accumulation, transport, dispersion of ozone.

### 3 Surface ozone concentrations affected by landfalling typhoons

      Fig.1 shows the tracks and categories of the 17 landfalling typhoons during the period of 2014–2018. Typhoons are

marked with 4 categories ranging from tropical storm to typhoons, severe typhoons and super typhoons. A total of 9 landfalling typhoons developed into the super typhoon intensity, 4 of which persisted their severity beyond the 24-h warning line when they moved westward. Typhoons landed on Taiwan (TW), Fujian (FJ) and Guangdong (GD) provinces most frequently, and kept travelling to influence neighbouring inland regions. Given a typical size of typhoon circulations at an order of 1000 km, these landfalling typhoons have the potential to influence large-scale meteorological environments and hence impact surface

ozone concentrations over southern China. Previous studies show that surface ozone concentrations in PRD and YRD are enhanced when typhoons are 700-1000 km away and when typhoons cross the 24-h warning line (thick black dashed line in Fig. 1) (e.g., Huang et al., 2006; Zhan et al., 2020). In addition to these valuable findings of ozone enhancement over the developed regions, this paper examines the overall surface ozone behaviours associated with landfalling typhoons over southern China, and comprehensively addresses both positive and negative impacts of typhoons on surface ozone

concentrations in the following part.

      Fig.2 shows surface ozone concentrations varying along the radial distance of typhoons at different typhoon developmental stages, based on the mean of the 17 landfalling typhoon samples. During the daytime (1400 LST, Fig. 2a), the background ozone concentrations range between 45-55 ppbv with small gradients along the radial direction of typhoons. Surface ozone concentrations are obviously elevated from the pre-typhoon conditions ($T_g$-2d and $T_g$-1d) to the typhoon genesis

($T_g$ and $T_g$+1d). The averaged ozone enhancement reaches 10-20 km when typhoons are 400-1500 km away, which produces a positive anomaly of 20-40% from $T_g$-2d to $T_g$+1d relative to the background ozone concentrations. Particularly, surface ozone concentrations are largest within radial distances of 1100-1300 km with a mean magnitude of 83.4 ppbv during $T_g$-1d and $T_g$ stages, yielding a positive anomaly of 52.5% against the background ozone concentrations of 54.7 ppbv. During the nighttime (0200 LST, Fig. 2b) when the ozone photochemical reactions cease, a positive ozone anomaly of ~9 ppbv relative

to the background ozone concentrations persists during $T_g$-1d and $T_g$ stages. The zone with significant ozone enhancement is





generally similar to that during the daytime. Considering the necessity of ozone episode warning, based on the consistent behaviours of ozone enhancement at daytime and nighttime behind a large ensemble of landfalling typhoons, we suggest that surface ozone concentrations at a radial distance of 1100-1300 km are most significantly enhanced during the initial stages of typhoons.

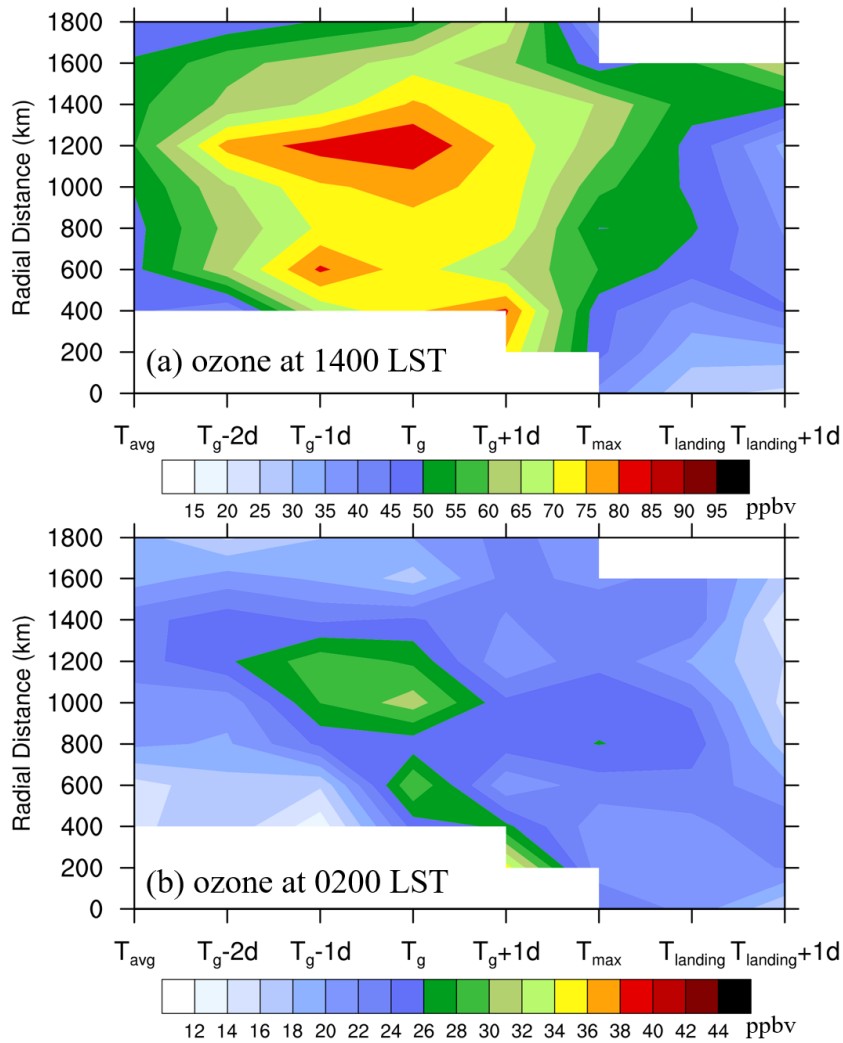


**Figure 2: Surface ozone concentrations (unit: ppbv) varying along the radial distance of typhoons at various typhoon developmental stages at daytime (a) and nighttime (b). The concentrations are the mean ozone concentrations of all the typhoons over 2014-2018. The blank at the bottom left and top right parts are due to missing observations.**

As the typhoons develop rapidly and approach southern China westward, the typhoon influences on surface ozone switch
from enhancement to reduction. An obvious ozone cleansing episode takes place right after the $T_{max}$ stage when typhoons have reached their maximum intensity and move closer to the coastal regions. Surface ozone concentrations fall into its background





levels at $T_{max}$ stage and keep decreasing when typhoons make landfall in southern China. Quantitatively, the negative ozone anomaly is -11.9 % at $T_{landing}$ and -16.6 % at $T_{landing}$ +1d stage relative to the background ozone concentrations at daytime. In addition, from $T_{max}$ to $T_{landing}$ +1d stage, it is clear that surface ozone concentrations increase monotonously with the radial
distance to typhoons increasing. In other words, partly due to the arrival of marine airmass, the ozone is greatly reduced near the typhoon centers.

To understand large changes in typhoon's influences on surface ozone from enhancing to cleansing ozone during the lifespan of typhoons, we explore meteorological connections between the evolutionary typhoons and successive response of surface ozone. Meteorological conditions that influence production, accumulation, transport and dispersion of ozone are
analyzed. Similar to the analysis applied to surface ozone, meteorological variables are processed with consideration of radial distance and developmental stages of landfalling typhoons. Fig. 3 shows the evolutions of air temperature and wind speed within ABL, and 500-hPa vertical air motions averaged over the 17 typhoons using the MERRA-2 reanalysis. From $T_g$-2d to $T_g$+1d stages, a systematic increase of air temperature is noticed both in near-surface 10-m and 850-hPa height within a radial distance of 400-1500 km to typhoon centers at daytime (Fig. 3a). Taking the air temperature as a proxy of solar radiation
intensity, it can be inferred that the boundary layer is dominated by fine and hot weather accompanied with strong solar radiation that promotes the photochemical production of ozone during these stages. In terms of wind fields, belt-like regions with weak winds (< 3-4 m/s) are found when typhoons are 800-1600 km away from $T_g$-2d to $T_g$+1d stages (Fig. 3b). The low wind speed zone extends up from surface to 850 hPa, yielding a stable ABL that is favourable for accumulation of ozone. Regarding vertical flows, downward air motions dominate in the mid troposphere over the outskirts of typhoons (800-1500
km in radial direction). This peripheral subsidence of typhoons contributes to the cloudless conditions and a stable structure of ABL, which are favourable to ozone production and accumulation. The meteorological scenario at nighttime is similar to that at daytime that support the persistency of promoted ozone contents except that the ozone photochemical reaction ceases due to lack of sunshine. During the early initial stages of typhoons, stagnation with low wind speed under the control of typhoons is also significant and accompanied by the systematic downward motions between 800-1200 km in radial direction
at night (0200 LST) as shown in Fig. 4.





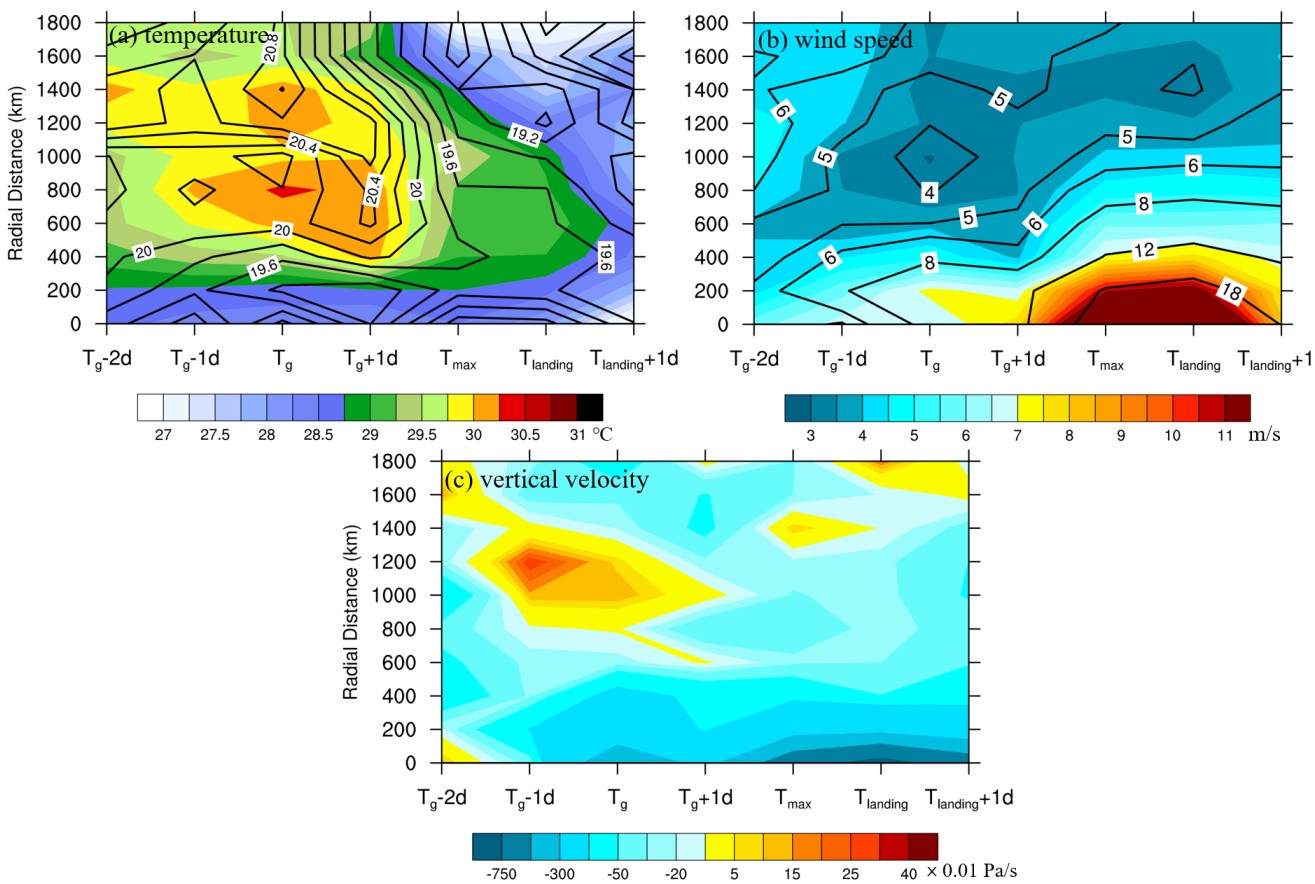

**Figure 3: (a) 10-m (shaded) and 850-hPa (contour lines) air temperature (unit: °C), (b) 10-m (shaded) and 850-hPa (contour lines) wind speed (unit: m/s) and (c) 500-hPa vertical wind velocity (unit: 0.01 Pa/s, positive values for downward air motions and vice versa) at daytime (1400 LST) based on MERRA-2 reanalysis at different radial distances and typhoon developmental stages averaged over the large ensemble of typhoons over 2014-2018.**

Along with the westward advance of typhoons, the weather begins to deteriorate and surface ozone concentrations drop after $T_g$+1d stage. The cloudy environments and convective activities take over the previous fine and hot weather. The air temperature drops significantly and the wind speed increases steadily from $T_{max}$ to $T_{landing}$ +1d stages that reduce accumulation of surface ozone (Fig. 3 and Fig. 4). As the typhoons approaching southern China, gales (wind speed > 10 m/s) appear and bring in the ozone-poor marine airmass. The upward vertical motions intensify over land and give rise to cloud formation and precipitation, which further reduce ozone concentrations.

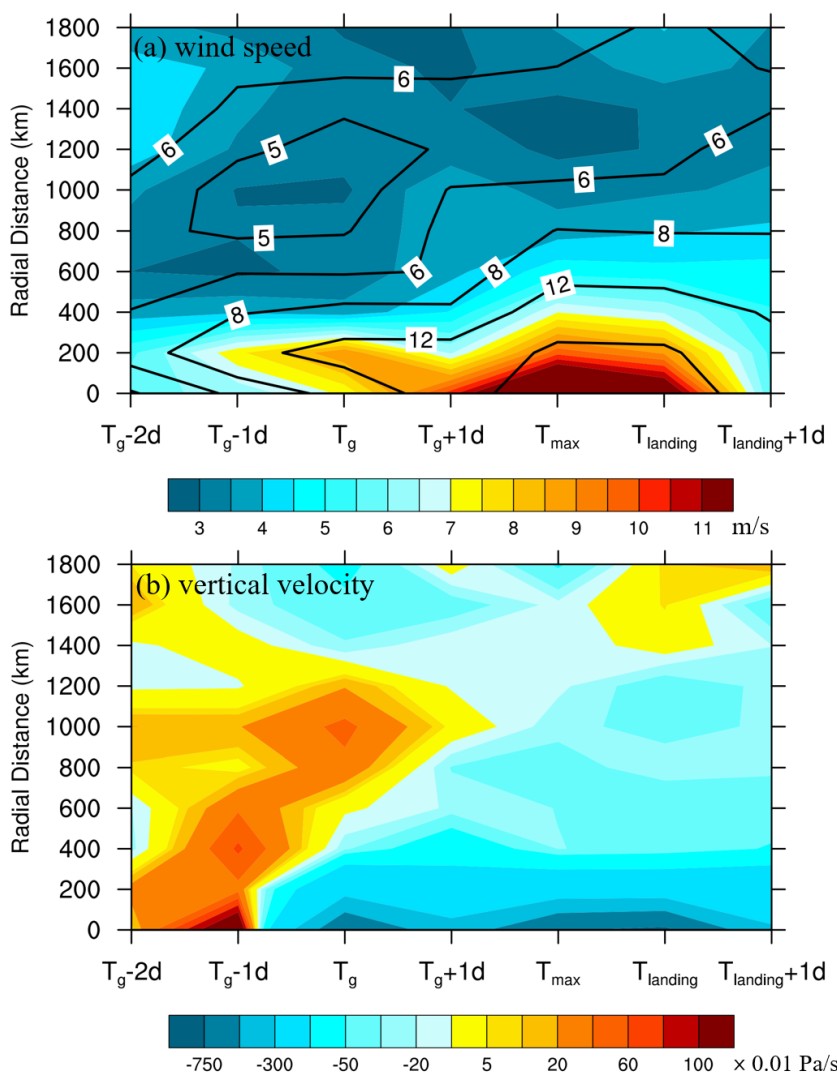

**Figure 4: Same as Fig. 3, but for nighttime (0200 LST).**

The above analysis presents the physical linkages between landfalling typhoons, meteorological conditions and surface
ozone variations. The overall meteorological conditions associated with different typhoon developmental stages alter
production, accumulation, transport and dispersion of ozone and lead to different influences from enhancing to cleansing ozone
pollution. Practically, these results of the spatiotemporal variations of ozone when a typhoon is approaching raise challenging
demands for observation and numerical forecasting of typhoon development. The necessary information about the timing and
location of typhoon genesis as well as pre-typhoon conditions, are only available from reliable numerical forecasting. Therefore,
to capture the evolutionary ozone behaviours over southern China during typhoon landfalling, not only chemical aspects of
models are required to describe the reactions between atmospheric compositions, but also meteorological conditions in models





should be improved to forecast the developmental stages of typhoons. Note that the analysis in this section is mainly related to the typhoon influences on ozone production and loss within the ABL. Several studies suggest that typhoon-induced STEs can bring ozone-rich airmass from the lower stratosphere and thus enhance surface ozone. Therefore, the variations of vertical
ozone profiles are analysed for assessing the typhoon influences on tropospheric ozone using temporally dense vertical observations in the next section.

## 4 Vertical ozone distributions affected by landfalling typhoons

It is reported that typhoons may enhance or reduce tropospheric or surface ozone concentrations (e.g., Jiang et al., 2015; Das et al., 2016; Li et al., 2020), which is probably related to the developmental stages and intensity changes of typhoons.
Therefore, dense vertical ozone observations are required given the rapid evolutions in intensity and location of typhoons. Using airborne observations of atmospheric compositions under the IAGOS framework, Roux et al. (2020) analysed the ozone profiles collected at Taoyuan International Airport (25.076 °N, 121.224 °E) of Taiwan (TW, Fig. 1) during the 2016 typhoon season. They found elevated ozone in the middle and upper troposphere and tracked back to its stratospheric origin. In this study, the ozone vertical profiles measured during taking-off and landing of flights at Taoyuan airport with dense temporal
resolutions are collected and processed according to the developmental stages of the typhoon ensemble. A total of 234 ozone profiles covering each divided stage of typhoons guarantee that the evolutionary variation in ozone profiles in response to typhoon developments can be well captured.

The averaged vertical ozone profiles at each typhoon developmental stage are shown in Fig. 5a. At $T_g$-2d stage, ozone concentrations range between 30-45 ppbv below 4 km altitude and increase slightly to 45-60 ppbv in the middle-to-upper
troposphere. Starting from $T_g$-1d stage, ozone concentrations in all layers increase with different magnitudes. Vertically, there appear two separated regions of high ozone abundances, one in the middle ABL and the other in the middle-to-upper troposphere. The high ozone abundances in the troposphere mainly have two sources (Zhan et al., 2020), e.g., active ozone photochemical reactions at daytime in the boundary layer and downward intrusions of ozone-rich air from upper levels. The dense airborne observations show that ozone enhancement below 2 km altitude is only significant in the pre-typhoon and initial
stages ($T_g$-1d and $T_g$), suggesting the dependence of photochemical reactions on meteorological conditions controlled by typhoons. Simultaneous measurements of temperature and winds via the flights (Fig. 6a and 6b) show that a peak positive temperature anomaly of 0.3 °C is located at 1.5 km altitude during $T_g$-1d and $T_g$ compared against that of $T_g$-2d stage, which is accompanied by a weak wind zone (< 6 m/s) promoting ozone production and accumulation in the boundary layer. Though the warming continues in the low troposphere due to the approaching warm core of typhoons, wind flows intensify rapidly in
the ABL and exceed 10 m/s that effectively transports clean air mass and lead to lower ozone concentrations after $T_g$+1d stage. In the middle-to-upper troposphere, ozone is enhanced and thus a region of high ozone concentrations of 75-80 ppbv appears at an altitude of 12 km at $T_{max}$ stage. As shown in Fig. 6c, the overall vertical air flows shift from weak subsidence to intensifying upward motions that perturbate the structure of tropopause. The decreasing tropopause might facilitate the





stratosphere-to-troposphere exchange and provide chances of ozone-rich air intrusions from the lower stratosphere. This may
explain an asynchronous evolution of ozone in the boundary layer and middle-to-upper troposphere.

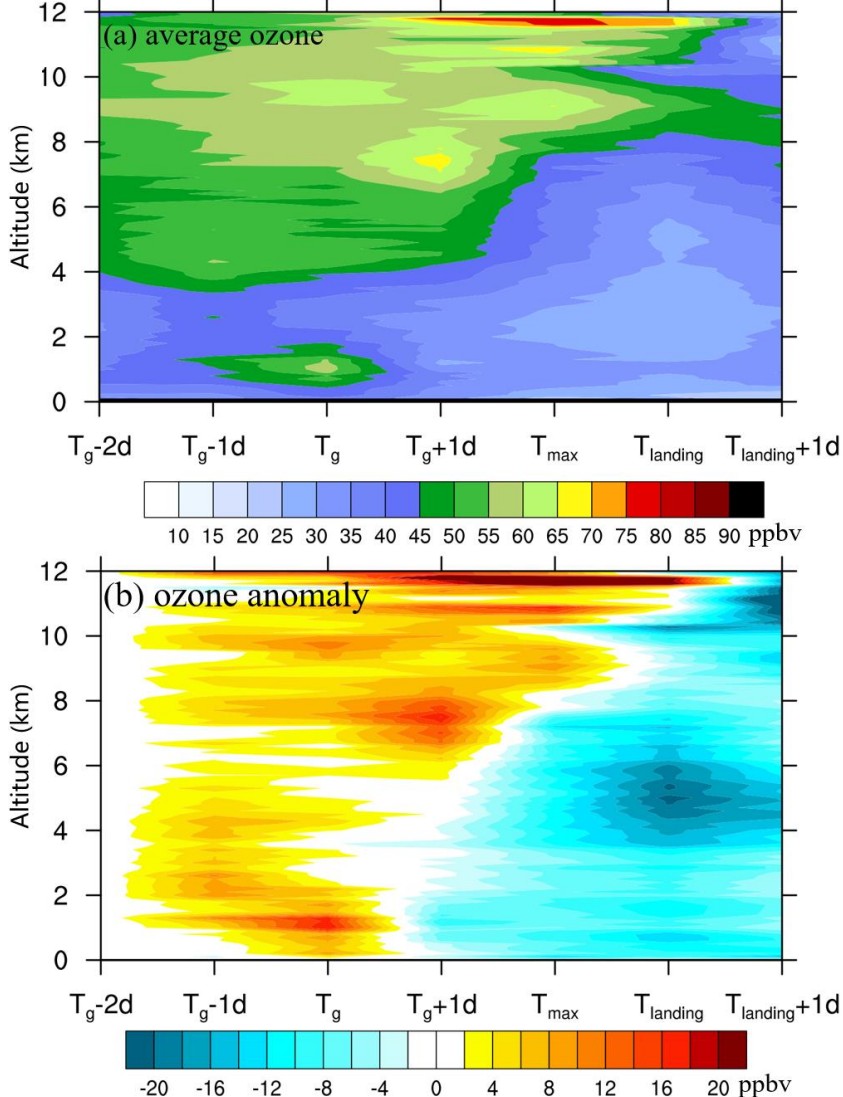

**Figure 5: (a) Vertical ozone concentrations (unit: ppbv) at each typhoon developmental stage averaged over the typhoon ensemble over 2014-2018 at Taiwan. (b) Anomaly of vertical ozone concentrations relative to Tg-2d stage of the subsequent typhoon developmental stages. The temporally dense ozone profiles are available from the airborne observations of atmospheric compositions under the IAGOS framework collected at Taoyuan International Airport of Taiwan.**

Taking the ozone concentrations at $T_g$-2d as the references, a tilted structure of ozone anomaly is obvious (Fig. 5b) due to the asynchronous evolution in different heights. Quantitatively, the maximum magnitude of ozone variations is 10-12 ppbv at 1-1.5 km altitude, equalling to a 25 % positive ozone anomaly at $T_g$ stage in the boundary layer. The results are consistent





with Zhan et al. (2020) who suggested that ozone is mainly generated inside the boundary layer (~ 1 km) instead of at the

surface. In the middle-to-upper troposphere, positive ozone anomalies persist over long time within a deep layer (4-12 km). The positive ozone anomaly reaches 10 ppbv at 7-8 km altitude at $T_g$+1d stage and 30 ppbv near 12 km altitude at $T_{max}$ stage. However, along with typhoon development, a negative ozone anomaly forms and stretches upward from $T_g$+1d stage, which probably is a compromise between upward transport of clean marine airmass and downward transport of ozone-rich air from upper levels. Hence, there appear largest negative differences of 14-18 ppbv around 5 km altitude at $T_{landing}$ stage and of 20-25

ppbv around 11 km altitude at $T_{landing}$+1d stage relative to ozone concentrations at $T_g$-2d stage.

Figure 6: (a) Temperature anomaly relative to $T_g$-2d (unit: °C) and (b) wind speed at each typhoon developmental stage within the boundary layer using the simultaneous flight observations at Taoyuan International Airport of Taiwan. (c) Vertical velocity (shaded, unit: 0.01 Pa/s) and the average height of 2-PVU (potential vorticity units, 1 PVU = 10−6 m2 s−1 K kg−1) and 3-PVU regions. The

data are averaged over the large ensemble of typhoons over 2014-2018.

The above analysis is based on the observational mean of eight typhoons over 2014-2018, suggesting that tropospheric ozone concentrations are reduced by clean marine airmass caused by the strong uplift in landfalling typhoons. Another question is how frequently the ozone-rich airmass from the upper levels can sink down to the lower troposphere and even enhance





surface ozone during the passing of typhoons. Reported in case studies by Das et al. (2016), the typhoon-induced downward
propagation of airmass can bring high ozone from the upper to the lower troposphere. Considering that such downward
transport of ozone-rich air mass could be easily smoothed by averaging all samples (e.g., Fig 5), we reexamine these temporally
dense ozone observations by counting the number of ozone concentrations (every 10 ppbv) in each vertical layer (with a 100-
m spacing) at different developmental stages of typhoons (Fig. 7). As mentioned above, a combination of high temperature
and weak winds in the boundary layer promotes the active ozone photochemical production and accumulation at $T_g$-1d and $T_g$
stages. Large ozone concentrations up to 130–160 ppbv below 2 km altitude are observed (black arrows in Fig. 7b-c). After
$T_g$+1d stage, the ozone episodes in the boundary layer cease due to deteriorated weather, however, the number of high ozone
concentrations grows in the upper troposphere. Using 80 ppbv as a threshold for ozone with stratospheric origin, based on the
averaged ozone concentrations in Fig. 5a, we found that the intrusions of stratospheric ozone-rich air largely appear above 4
km altitude (red arrows in Fig. 7d-f). Despite some mixing processes with ambient air, the stratospheric air mass can also sink
down to the lower boundary layer (below 2 km altitude). As shown in the black circles in Fig. 7d-f, the number of ozone
concentrations over 60 ppbv rises at $T_{max}$ stage compared with those at $T_g$+1d and $T_{landing}$ stages, which evidently confirms the
contribution of external stratospheric ozone to lower troposphere ozone. With a quantitative overview of the vertical ozone
variations in the eight typhoons, we found that five of them deduce stratospheric intrusions and the ozone-rich stratospheric
airmass sink down to the ABL observed in 3 typhoon cases. Though ozone concentrations are reduced in the entire tropospheric
column vertically when typhoons are making landfalling, surface ozone is possibly enhanced by the downward propagation of
stratospheric ozone-rich air.







**Figure 7: Number of ozone concentrations (every 10 ppbv) in each vertical layer (with a 100-m spacing) at different developmental stages of typhoons. The red dashed lines of 80 ppbv represent ozone-rich air mass with stratosphere origin based on the averaged**
**results in Fig. 5a.**

## 5 Conclusions, discussions and suggestions

In previous studies, influences of typhoons on ozone in the atmospheric boundary layer were investigated over the developed and populated regions, such as PRD and YRD. Given the severity of ozone pollution during rapid urbanization and economic development in southern China, it is of importance to address and quantify impacts of typhoons on ozone over larger
domains that are frequently under controls of typhoons. In this study, we use an ensemble of 17 landfalling typhoons over 2014-2018 (Fig. 1) to investigate the positive and negative influences of typhoons on tropospheric ozone over southern China. Both proximity to typhoons and typhoon developmental stage are taken into account to reveal the evolutionary response of





tropospheric ozone to landfalling typhoons. We found that surface ozone is enhanced when typhoons are 400-1500 km away during the initial stages of typhoons (e.g., from 1 day before and to 1 day after typhoon genesis). On average, the positive ozone anomaly reaches 10-20 ppbv at the daytime (1400 LST) and 9 ppbv at nighttime compared with the background ozone level. Surface ozone concentrations at radial distances of 1100-1300 km are most significantly enhanced during the initial stages of typhoons. As the typhoons move closer to southern China westward, the influences of typhoons change from enhancing to reducing ozone. Then, typhoons reach their maximum intensity and keep decreasing in their intensity when they make landfall, and surface ozone concentrations are reduced with a negative ozone anomaly ranging between -12 % ~ -17 % relative to the background ozone level.   The physical linkages between typhoons, meteorological conditions and ozone are investigated. Results show that a combination of increasing air temperature, weak winds in the ABL and dominating downward motions promotes the photochemical production and accumulation processes of ozone over the outskirts of typhoons during their initial stages. When typhoons are making landfalling, the deteriorating weather accompanied by dropping temperature and wind gales reduces the production and accumulation of surface ozone. Ozone-poor marine airmass are brought to inland. In addition, the intensified upward vertical motions give rise to cloud formation and precipitation and further hinder ozone formation and accumulation.

Besides the processes in the ABL influenced by typhoons, we also investigate variations in tropospheric ozone profiles during the differential developmental stages of landfalling typhoons. In particular, we examine how the typhoon-induced stratospheric intrusions alter lower troposphere and surface ozone. Based on the temporally dense ozone vertical observations collected at Taiwan during eight typhoons, we found two regions of high ozone abundances separately located in the ABL and the middle-to-upper troposphere. In the ABL at the initial stages of typhoons, ozone below 2 km altitude is generally enhanced because of the warming air and relatively low wind speed. Then ozone concentrations decrease continuously when wind intensifies rapidly that transport the clean air mass. In the middle-to-upper troposphere, ozone enhancement persists over a long period generating a region of high ozone concentrations (75-80 ppbv at 12 km altitude). The tropopause is perturbated with decreasing tropopause height as typhoons develop, which might provide many chances of stratospheric intrusions that bring ozone-rich air from the lower stratosphere. The asynchronous evolutions of ozone in the ABL and middle-to-upper troposphere lead to a tilted structure of ozone anomaly vertically when typhoons evolve (Fig. 5). Averagely, the positive ozone anomaly maximizes around 10-12 ppbv at 1-1.5 km altitude at the initial stages of typhoons. In the middle-to-upper troposphere, the positive ozone anomaly is 10 ppbv at 7-8 km altitude shortly after typhoon genesis, and 30 ppbv near 12 km altitude when typhoons reach their maximum intensity. When typhoons are landing, the negative ozone anomaly stretches upward with a maximum ozone reduction of 14-18 ppbv at 5 km altitude and 20-25 ppbv at 11 km altitude relative to the pre-typhoon conditions.

We further assess the impacts of typhoon-induced disturbances in the upper troposphere that bring ozone-rich air downward at different developmental stages of typhoons (Fig. 7). During the initial typhoon stages, high ozone (130–160 ppbv) below 2 km altitude are observed, which is attributed to active ozone photochemical production and accumulation processes. After typhoon genesis, ozone-rich air from the stratosphere is more frequently observed above 4 km altitude, and even



propagates downward to the lower boundary layer (below 2 km altitude) despite some mixing processes with ambient air. We found that in the eight typhoons with ozone profile observations covering the entire typhoon lifespan, five of them deduce ozone-rich air with the stratospheric origin, and in 3 typhoon cases the intrusions can penetrate down to the ABL. This suggests

that surface ozone is possibly enhanced by the downward propagation of stratospheric ozone-rich air when typhoons reach their maximum intensity, though the tropospheric column ozone is usually reduced after landings of typhoons.

Using a large ensemble of typhoons and temporally dense ozone observations, this study characterizes spatiotemporal variations in tropospheric ozone under the influence of typhoons over southern China, a region with frequent typhoon activities, and investigate the positive and negative influences of typhoons on surface ozone and vertical variations in tropospheric ozone.

The impact of typhoon-induced stratospheric intrusions is quantitatively examined to reveal the possibility of surface ozone episodes suffering from upper-level ozone sources. Still, further studies are needed to better assess the contributions of different chemical and physical processes to ozone concentrations in the ABL and to assess the role of STEs in increasing tropospheric and surface ozone during typhoons. Intensive observations of vertical ozone profiles are highly demanded over different typhoon regimes, and hence we plan to conduct ozonesonde measurements at Fujian province that complement airborne ozone

observations at Taiwan, both of which are frequent landing spots of typhoons. Numerical simulations of meteorological and chemical evolutions during typhoons, for example, using the Weather Research and Forecasting (WRF) model coupled with Chemistry (WRF-Chem), provide a way to analyse ozone variations at fine scale. However, both the meteorological and chemical simulations need to be improved so the evolutionary features of ozone during typhoon landfalling can be captured. Also, previous studies stressed the lightning associated with intensive convection of typhoons can produce nitrogen oxides

($LNO_x$) and hence influence ozone chemical reactions (Kaynak et al., 2008; Roux et al., 2020; Das et al., 2016). In addition to lightning occurrences and $LNO_x$, the deep convection of typhoons can also transport ozone by dynamically dragging down the stratospheric ozone-rich air. Pan et al. (2014) reported that the ozone-rich stratospheric air wraps around both leading and trailing edges of a mesoscale convective system and descends to lower levels. Hence a better representation of the $LNO_x$ influence on chemical reactions and meteorology concerning dynamical transport should be included in the typhoon

simulations. Currently, we are incorporating data assimilation (DA) to improve WRF-Chem simulations. We are developing a three-dimensional variational DA scheme to assimilate lightning observations over the data-sparse oceans to update meteorological conditions (Chen et al., 2020) and $LNO_x$ presentations (Allen et al., 2010; Pickering et al., 2016; Kang et al., 2019).

**Data Availability Statement**

The track data of tropical cyclones for China used in the present study can be obtained from http://tcdata.typhoon.org.cn/ zljlsjj_sm.html. The surface air pollutant observations obtained from the China National Environmental Monitoring Centre can be obtained from http://106.37.208.233:20035/. The airborne measurements of atmospheric chemical compounds provided



by the European Research Infrastructure program IAGOS can be downloaded from https://www.iagos.org. The MERRA-2 reanalysis meteorological data can be downloaded from https://gmao.gsfc.nasa.gov/reanalysis/MERRA-2.

**Acknowledgments**

The computing resources used in this study were provided by Fujian Normal University High Performance Computation Center (FNU-HPCC). We also acknowledge the free use of flight-based atmospheric chemical measurements from IAGOS. IAGOS is funded by the European Union projects IAGOS-DS and IAGOS-ERI. The IAGOS database is The MOZAIC/CARIBIC/IAGOS data were created with support from the European Commission, national agencies in Germany
(BMBF), France (MESR), and the UK (NERC), and the IAGOS member institutions (http://www.iagos.org/partners). The participating airlines (Lufthansa, Air France, Austrian, China Airlines, Iberia, Cathay Pacific, Air Namibia, Sabena) supported IAGOS by carrying the measurement equipment free of charge since 1994.

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
