# Peer review of "Positive and negative influences of landfalling typhoons on tropospheric ozone over southern China"

_Atmospheric Chemistry and Physics, 2021_

## Author Response (AR1)

**Responses to reviewers' comments**

Dear Editor,

We are very pleased to submit the revised manuscript entitled "Positive and negative influences of landfalling typhoons on tropospheric ozone over southern China" for possible publication in journal of ACP.

I'd like to thank you for your attention for this matter. The reviewers' constructive comments and suggestions are greatly appreciated. The paper has been revised according to the reviewers' recommendations. Below please find our detailed point-by-point responses (in red) to the reviewers' comments (in black) to the manuscript.

Yours sincerely,

Zhixiong Chen

On behalf of all authors

**Review#1**

Please concise the abstract.

We greatly appreciate the thorough review and helpful suggestions provided by the referee. The abstract has been revised in the manuscript.

**Review#2**

The authors present that 17 landfalling typhoons impact on surface ozone and tropospheric ozone over southern China. Using the MERRA-2 reanalysis data, IAGOS and ground-based ozone measurements, the authors report that surface ozone is enhanced by 9-20 ppbv and tropospheric ozone below 12 km enhanced at radial distances of 400-1500 km away during the initial stages of typhoons. When typhoons are landing, surface ozone decreases and tropospheric ozone reduces by 14-25 ppbv.

Impact of such typhoons on ozone is welcome and, indeed, it is important to characterize the physical linkages between typhoons, meteorological conditions, and ozone variations. In that respect, the datasets used are appropriate, the texts are well organized, and English need to be improved. Overall, the manuscript well fits into the ACP scope and I recommend its publication after revisions.

We thank the anonymous referee for his/her insightful and constructive comments.

Below are our point-to-point responses in detail.

Specific comments:

Line18: …ranging between -12 % ~ -17 % relative to the background ozone level. How about using ppbv for ozone decrease?

Thank you for this suggestion. The negative ozone anomaly relative to background level was about 6-9 ppbv. We have revised the expression according to your suggestion.

Line44-45: Line 55-58: … in right format.

Corrected.

Line53: …wind speeds exceeding 37.2 m/s. →…wind speeds exceeding 37.2 m s$^{-1}$.

Corrected.

Line76: …the role of typhoons in cleansing the air and reducing…→…the role of typhoons in cleaning the air and reducing…

Corrected.

Line79: Nara → Naha

Sorry for this typo. We corrected it in the manuscript.

Line113-116: m/s →m s$^{-1}$

Corrected.

Line165: For O3→For O$_3$

Corrected.

Line190-192: A total of 9 landfalling typhoons developed into the super typhoon intensity, 4 of which persisted their severity beyond the 24-h warning line when they moved westward. Fig. 1 shows the tracks of the 17 landfalling typhoons. Here, the authors show 9 landfalling typhoons. Not clear, the left of the 17 typhoons?

Actually, the results concerning the surface ozone concentrations are the ensemble mean of the 17 landfalling typhoon cases during 2014-2018. We check the original typhoon track data again, and found that among the 17 typhoons, 11 of them had developed into the super typhoon (Super TY) category. Sorry for this mistake, and we have revised the descriptions in the manuscript.

Line215: for Figure 2, the surface data from the China National Environmental Monitoring Centre? Add the data source.

Yes, the data were downloaded from China National Environmental Monitoring Centre. Data source was added in Line 215.

Line258: for Figure 4, Same as Fig. 3b-c...

Corrected.

Line293, Line295: m/s →m s$^{-1}$

Corrected and checked throughout the manuscript.

Line317: for Fig. 6c. recommend to add the temperature lapse-rate tropopause.

Thank you so much for this valuable suggestion. Besides the dynamical PV-based tropopause height, we have also tried to calculated the temperature lapse-rate tropopause. In the calculation of temperature lapse-rate tropopause, vertical profiles of temperature are necessary to examine the fine scale variations of temperature. However, given the coarse vertical resolution of MERRA2 reanalysis data (approximately 1 km vertical resolution between 10-16 km altitude), it is often hard to determine the height of temperature lapse-rate tropopause. Facing this problem, we checked the GPS radio occultation data from the Constellation Observing System for Meteorology, Ionosphere, and Climate (COSMIC), which detect vertical thermal structures in strong convective systems with better vertical resolution. It is a pity that COSMIC provided limited number of temperature profiles over West Pacific Ocean in the typhoon cases, and hence difficult to offer reliable estimation of temperature lapse-rate tropopause variations. As a result, only the PV-based tropopause height is provided in the manuscript.